# A Flow Cytometry-Based Examination of the Mouse White Blood Cell Differential in the Context of Age and Sex

**DOI:** 10.3390/cells13181583

**Published:** 2024-09-20

**Authors:** Elise Arlt, Andrea Kindermann, Anne-Kristin Fritsche, Alexander Navarrete Santos, Heike Kielstein, Ivonne Bazwinsky-Wutschke

**Affiliations:** 1Institute of Anatomy and Cell Biology, Medical Faculty, Martin-Luther-University Halle-Wittenberg, 06108 Halle (Saale), Germany; andrea.kindermann@uk-halle.de (A.K.); anne-kristin.fritsche@medizin.uni-leipzig.de (A.-K.F.); heike.kielstein@uk-halle.de (H.K.); ivonne.bazwinsky@uk-halle.de (I.B.-W.); 2Institute of Anatomy, Medical Faculty, University of Leipzig, 04103 Leipzig, Germany; 3Core Facility Flow Cytometry, Center for Basic Medical Research, Medical Faculty, Martin-Luther-University Halle-Wittenberg, 06108 Halle (Saale), Germany; alexander.navarrete@uk-halle.de

**Keywords:** immune cells, blood, leukocytes, leukocyte subsets, mouse, reference values, blood sampling, immune system, mouse blood, mouse blood cells

## Abstract

Analysis of the white blood cell differential as part of a flow cytometry-based approach is a common routine diagnostic tool used in clinics and research. For human blood, the methodological approach, suitable markers, and gating strategies are well-established. However, there is a lack of information regarding the mouse blood count. In this article, we deliver a fast and easy protocol for reprocessing mouse blood for the purpose of flow cytometric analysis, as well as suitable markers and gating strategies. We also present two possible applications: for the analysis of the whole blood count, with blood from a cardiac puncture, and for the analysis of a certain leukocyte subset at multiple time points in the framework of a mouse experiment, using blood from the facial vein. Additionally, we provide orientation values by applying the method to 3-month-old and 24-month-old male and female *C57BL/6J* mice. Our analyses demonstrate differences in the leukocyte fractions depending on age and sex. We discuss the influencing factors and limitations that can affect the results and that, therefore, need to be considered when applying this method. The present study fills the gap in the knowledge related to the rare information on flow cytometric analysis of mouse blood and, thus, lays the foundation for further investigations in this area.

## 1. Introduction

The analysis of the white blood differential is a widely used diagnostic procedure in medical practice and research. It offers valuable insights into the immune system’s status and provides essential information about an organism’s overall health. The traditional method involves the differentiation of leukocytes in a blood smear under a microscope and analyzing the proportions of lymphocytes, monocytes, neutrophils, eosinophils, and basophils [1]. Today, plenty of hematology devices automate this process using flow cytometry-based methods. However, those devices are adapted to human blood and cannot be easily transferred to animal models, due to species-dependent differences. In addition, without prior labeling of the cells, lymphocyte populations cannot be further differentiated into B or T lymphocytes. They are all similar in size, appearance, and histological stainability [1]. Fluorochrome-associated, flow cytometry-based analyses of human peripheral blood leukocytes are often performed in medical research and diagnosis, since several diseases are followed by an immune reaction or may even originate from the immune system [2]. The methodological approach, suitable markers for specific leukocyte populations, gating strategies, and ways to avoid frequent problems, are well-established for human blood [3,4].

Mice are the most frequently used mammals in research. In Germany, they constituted 72.4% of all vertebrates in 2022 [5]. Still, very little information exists about the flow cytometry-based blood count of mice, which might be due to the small amount of blood that can be harvested and the resulting challenge of gaining PBMCs. Cossarizza et al. have conducted a very detailed phenotyping of human and mouse leukocytes, including various markers and protocols, as well as guidelines to follow for flow cytometric analysis [6]. Skordos et al. provide a protocol for the analysis of mouse T cell subsets and analyze them in different immune cell compartments [7]. Liu et al. describe the gating of myeloid leukocyte subsets in different mouse tissues [8]. Despite many different publications on partial aspects of the topic, we are not aware of any studies in the literature that deal comprehensively and exclusively with mouse blood analyses and that contain achievable results. Therefore, this paper offers a comprehensive overview of the subject and advocates for the use of standardized flow cytometry-based analyses for mouse blood.

The immune system of mammals consists of the innate and the adaptive immune system. Cells in the innate immune system are granulocytes, monocytes, macrophages, natural killer (NK) cells, mast cells, dendritic cells, and antigen-presenting B lymphocytes. The adaptive immune system mainly consists of T lymphocytes and antibody-producing B lymphocytes [9]. Even though all of these cell populations can be found in both humans and mice, there are differences concerning the surface markers and proportions [10]. One of the biggest differences is that humans, as well as horses, dogs, and cats, have a granulocytic blood count. This implies that granulocytes constitute more than 50% of all blood leukocytes. In contrast, rodents (as well as ruminants and pigs) appear to have a lymphocytic blood count, with more than 50% of all leukocyte subsets being lymphocytes [11].

In addition, there are differences between the mouse strains. Over 450 inbred laboratory mouse strains have been described, providing a huge amount of different genotypes and phenotypes for multiple studies, and new strains are still generated regularly [12]. The most widely used mouse model for experimental studies and a common background for genetically modified mouse models are *C57BL/6* mice, due to the availability of congenic strains, easy breeding, and robustness [13].

Besides all these genetic factors, the composition of blood undergoes constant change [14]. Age, sex, gravidity, lactation, the feeding regime, environmental influences, and even stress, can have an impact on immune cell proportions in the blood [15,16,17,18]. In common, specific-pathogen-free laboratory animal management, external influences are nearly erased. Differences in sex, age, and stress, still play a role in experimental designs. In acute stress situations, the hypothalamic–pituitary–adrenal axis and the sympathetic nervous system are activated, which results in a release of stress hormones from the liver and stress-induced hyperglycemia [19,20]. Such factors must be considered when planning animal experiments.

This study focuses on gaining information about the status of different immune cell populations using only a small amount of blood. We offer a simple and reproducible method that can be used in any laboratory. We provide six distinct panels as well as gating strategies for an eight fluorochrome-based flow cytometry analysis of the whole mouse blood leukocyte subsets and give suggestions for additional markers. The panels can be utilized to create a comprehensive white blood cell differential, when using blood obtained from a cardiac puncture. However, they can also be applied separately, if there is a specific interest in analyzing a particular type of cell. In this case, the analysis can also be integrated into an ongoing experiment at several time points. Peripheral blood from the facial vein can be used to examine alterations in blood leukocytes during the progression of a disease. Additionally, we provide orientation values in the context of age and sex, by applying the method to 3-month-old and 24-month-old male and female, healthy, *C57BL/6J* mice.

## 2. Materials and Methods

### 2.1. Animals

All the animal experiments were approved by the local animal welfare officer. The criteria according to the 3Rs (Replacement, Reduction, and Refinement) were considered and applied accordingly. The *C57BL/6JRj* mice were purchased from Janvier Labs (Le Genest-Saint-Isle, France). After delivery, the mice were housed in a temperature- and humidity-controlled vivarium, with a 12 h light–dark cycle (L/D = 12:12, light on 06:00 a.m.) for a one-week acclimatization period and were supplied with food and water ad libitum, while being fed a standard diet. For the study, 24 mice were split into four groups: male young mice (3 months, *n* = 6, 3 Mo M), female young mice (3 months, *n* = 6, 3 Mo F), male aged mice (24 months, *n* = 6, 24 Mo M), and female aged mice (24 months, *n* = 6, 24 Mo F). The young and aged mice were each delivered together, and all the experiments were performed within 7 days, after the acclimatization period.

### 2.2. Reagents, Material, and Equipment

Every antibody used in this study was purchased from Miltenyi Biotec and was produced with REAfinity^TM^ technology, thus no Fc receptor block was needed and non-specific binding was eliminated [21]. Naturally, conventional monoclonal antibodies from other suppliers can also be used. Additionally, all of the used antibodies bind to surface markers, which simplifies the protocol by making permeabilization steps unnecessary. A list of the reagents, materials, and equipment used in this study can be found in the Appendix A.

### 2.3. Blood Collection and Reprocessing

All of the performed experiments took place in the morning, during the light period (8:00–11:00 a.m.). The mice were euthanized with CO_2_. The facial vein was then punctured with a lancet, and 3–5 drops (60–80 µL) of blood were obtained and collected in an EDTA-coated tube to prevent coagulation. The blood glucose levels were assessed by obtaining a small blood sample from the tail tip and analyzing it using a blood glucose monitoring device. A cardiac puncture was performed with a 3 mL syringe and an EDTA-coated cannula to prevent coagulation, followed by cervical dislocation. The collected blood samples had a volume of 700 µL (female young mice) to 1300 µL (male aged mice) and were transferred to an EDTA-coated tube to be stored at room temperature until further reprocessing. Other tissue samples and organs were harvested to be kept for analyses in further studies.

A total of 700 µL of blood taken via the cardiac puncture was transferred to a 15 mL tube and ten times the amount of red blood cell lysis buffer (7 mL) was added. The sample was incubated for 10 min on ice, then the same amount of FCA buffer (7 mL, flow cytometric analysis buffer) was added to stop the lysis process. The blood cell mixture was centrifuged for 5 min at 4 °C with 500× *g,* and the supernatant was discarded. The erythrocyte lysis was repeated once more. Then, 600 µL of FCA buffer was added to the tube and the cell suspension was split into six 5 mL tubes.

The reprocessing of the harvested blood from the facial vein was carried out in parallel, after the same protocol. Moreover, 50 µL of blood was transferred into a 5 mL tube and ten times the amount of red blood cell lysis buffer (500 µL) was added. The sample was incubated for 10 min on ice, then the same amount of FCA buffer (500 µL) was added to stop the lysis process. The blood cell mixture was centrifuged for 5 min at 4 °C with 500× *g*, and the supernatant was discarded. The erythrocyte lysis was repeated once more. Then, 100 µL of FCA buffer was added to the tube.

Then, 5 µL of the blood gained via the cardiac puncture was used to prepare a blood smear for every mouse. The dry slides were fixated with ethanol and stained with Pappenheim staining. Under a microscope, 130 cells were randomly counted and differentiated.

### 2.4. Antibody Staining and Analysis at the MacsQuant 16

A master mix was prepared in advance for each panel, by dissolving 0.3 µg of each antibody in 90 µL of FCA buffer. Then, 100 µL of the antibody master mix was added to 100 µL of the blood cell suspension and the suspension was incubated for 15 min at 4 °C in the dark. Then, 1 mL of FCA buffer was added and the tubes were centrifuged for 5 min at 4 °C with 500× *g*. The supernatant was discarded and the tubes were filled with 300 µL of FCA buffer. The samples were stored on ice in the dark until analysis, within 2 h. Right before the measurement at the MacsQuant 16, 2 µL of propidium iodide (PI) was added to the samples. The samples were measured completely or (if there were more) stopped at 150,000 counts. Compensation was carried out with anti-REA compensation beads and the MacsQuant auto-compensation tool. The compensation results were controlled and manually modified with the help of single staining, FMO controls (Appendix A), and unlabeled controls.

Table 1 shows the seven panels that were used in this study. Panels 1–6 were performed with blood from the cardiac punctures and the proportions of all the leukocytes circulating in the blood were precisely analyzed, including some important subsets. Panel 7 was performed with peripheral blood from the facial vein and the most important circulating immune cell subsets were roughly outlined.

### 2.5. Statistical Evaluation and the Handling of Outliers

The visualization and statistical evaluation of the data was performed with GraphPad Prism (Prism 10, GraphPad Software Inc., San Diego, CA, USA). To test the statistical significance between the groups, a one-way ANOVA test was performed. The *p* values are indicated in the graphs if they are smaller than 0.1 (90% confidence interval). The data are presented as the mean.

Three of the animals in the aged group appeared to have neoplastic alterations (two females, one male), which can happen occasionally at that age. These animals showed a few strongly deviating outliers, which were excluded from the analysis. All the outliers pertaining to the mice that appeared healthy during dissection were included.

## 3. Results

### 3.1. Gating Strategies

Figure 1 shows a basic classification of leukocyte subsets in the blood of mice and the expression pattern of the markers that are necessary for their clear identification. The following chapters include a detailed description of the markers and gating strategies for leukocyte subsets circulating in mouse blood. Additionally, we provide information about where to find specific leukocyte populations in the FSC/SSC and the CD45/SSC plots, so they can be consulted as support in unclear cases. All the gates were set, if unclear, in regard to the FMO controls with isotype antibodies (Appendix A). The flow cytometry results were analyzed using FlowJo™ v10.10Software (BD Life Sciences) [22]. Figure 2, Figure 3, Figure 4, Figure 5, Figure 6, Figure 7 and Figure 8 include representative plots for the young mouse group. Preliminary experiments have been incorporated to effectively illustrate the gating strategy and substantiate the claims that will be presented in the following chapters.

Figure 2 shows the initial gating steps. All of the following gating strategies continue to be used from this point onwards. Doublet exclusion is recommended, but there should be no more than 2% doublets in the sample (Figure 2A/2). Dead cells are excluded with PI or DAPI (Figure 2A/3) and accumulate as a small population in the FSC/SSC (Figure 2B/1). CD45 is a well-known, pan-leukocyte marker and should be used to eliminate the remaining erythrocytes, platelets, and non-blood cells (Figure 2A/4,B/2) [22].

The FSC/SSC plot is individual, shows a wide heterogeneity, and should be treated with caution. There is no clear separation between lymphocytes and monocytes, as is the case with human blood [23]. We can only distinguish between an SSC^low^ (mostly lymphocytes, monocytes, basophils), an SSC^int^ (mostly neutrophils), and an SSC^high^ (mostly eosinophils) population (Figure 2A/5, see the following chapters).

The CD45/SSC plot can be divided into five populations: a CD45^-^ population (erythrocytes, platelets, non-blood-cells), a small CD45^low^SSC^low^ population (mostly basophils), a CD45^int^SSC^int^ population (mostly neutrophils), a CD45^int^SSC^high^ population (mostly eosinophils and macrophages), and a CD45^high^SSC^low^ population (mostly lymphocytes, monocytes, Figure 2A/4, see Section 3.1.5). In some cases, gating with SSC might be helpful, but not using specific markers results in inaccuracies.

**Figure 1 cells-13-01583-f001:**
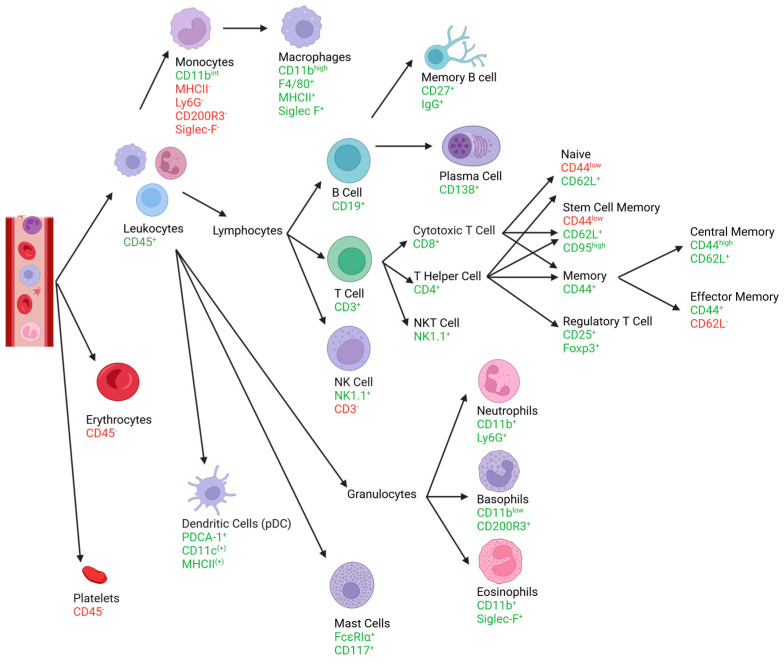
The graphic shows the most important leukocyte subsets and their surface markers for identifying them. For each cell type, the corresponding pathway can be followed, based upon which the expressed markers are listed. This figure was designed on the basis of Section 3.1 “Gating strategies”. Created with BioRender.com.

**Figure 2 cells-13-01583-f002:**
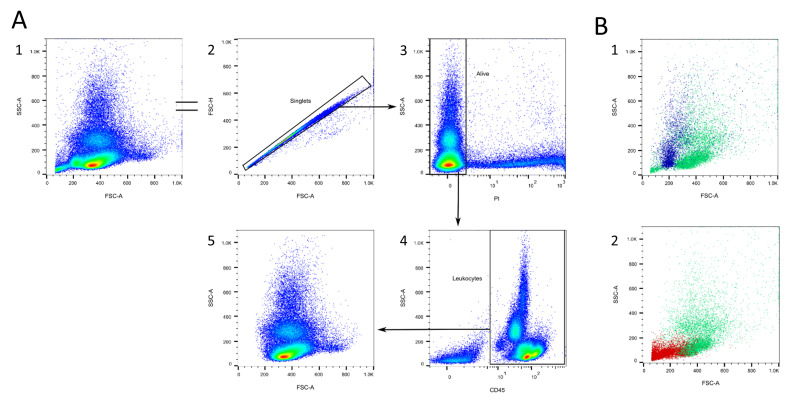
First gating steps. (**A**) The doublets excluded (2). Dead cells are excluded with the help of propidium iodide (PI, 3). Then, the CD45^+^ cells are selected (4). (**B**) Dead cells accumulate as a small population in the FSC/SSC plot (blue, 1). Erythrocytes and platelets accumulate in the corner of the FSC/SSC plot (red, 2).

#### 3.1.1. T Cells

The gating of T cells and their subtypes is exemplified in Figure 3. They are characterized by the expression of CD3 (Figure 3A/2) and can be further differentiated by their expression level in terms of CD4 and CD8 into CD4^+^ T cells (mainly T helper cells), CD8^+^ T cells (cytotoxic T cells), and CD4^−^CD8^−^ T cells (Figure 3A/3) [7,24,25]. Interestingly, CD4^+^ T cells have a higher density of CD3 than CD8^+^ T Cells (Figure 3B). Both CD4^+^ and CD8^+^ T cells can further be differentiated into four groups by their surface expression in terms of CD44, CD62L, and CD95 (Figure 3A/5–8). Both naïve and stem cell memory T cells express the lymph node homing molecule CD62L and low CD44. T_SCM_ are characterized by a high expression of the memory marker CD95. They display a naïve-like phenotype, but are minimally differentiated and bridge naïve and conventional memory T cells [26]. Out of the CD44^+^ population, CD44^high^CD62L^+^ central memory T cells and CD44^+^CD62L^−^ effector memory T cells can be identified [7,24]. If the interest is in lymphocyte activation, CD69 can be used as activation marker [27].

Out of the CD4^+^ population, CD25^+^FoxP3^+^ regulatory T cells can be identified (Figure 3A/4). CD25 is known to be a marker for regulatory T cells, but can also be upregulated in activated T lymphocytes [28,29]. Therefore, FoxP3 is more specific, but it is not used in this study as it is an intracellular transcription factor [24,30]. For staining FoxP3, additional permeabilization steps are necessary.

The stem cell antigen-1 (Sca-1, Ly6A) is a marker for hematopoietic progenitor cells that can differentiate into myeloid, B cell, and T cell lineages [31]. It is also strongly expressed in T_SCM_ cells [26]. Sca-1 is downregulated with increasing differentiation, but is then re-expressed on peripheral, mature circulating T cells and activated lymphocytes [32,33]. The expression is especially high during infections and remains high in virus-specific memory cells [32]. Both CD4^+^ and CD8^+^ T cells express different amounts of Sca-1 (Figure 3C).

T cells are located in the SSC^low^ population in the FSC/SSC plot (Figure 3D/1). They express a high amount of CD45 and are, therefore, located in the CD45^high^SSC^low^ population in the CD45 SSC plot (Figure 3D/2).

**Figure 3 cells-13-01583-f003:**
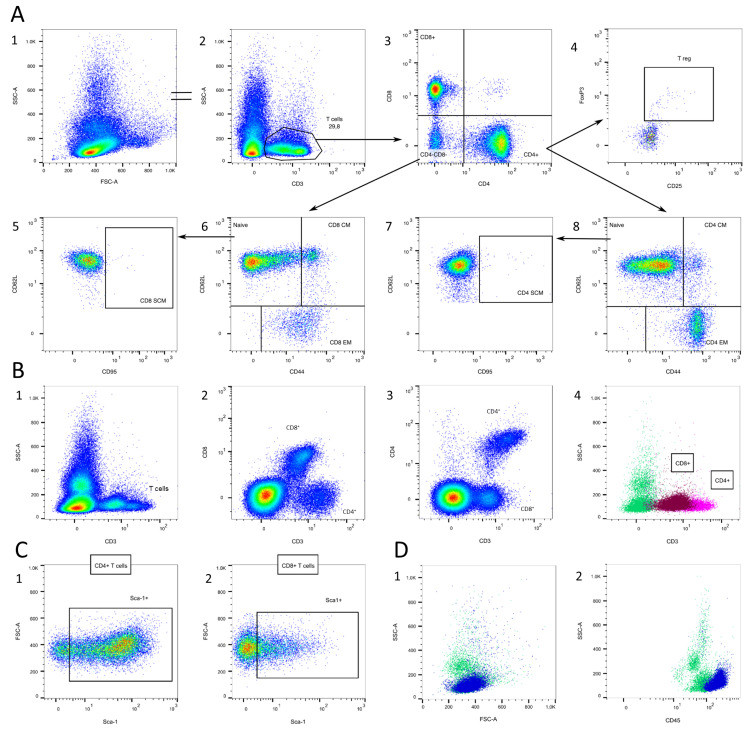
T cell gating. (**A**) CD3^+^ cells (T cells) are selected (2). T cells can further be classified by their expression in terms of CD4 and CD8 (3). CD4^+^CD25^+^FoxP3^+^ regulatory T cells can be defined (4). Both CD4^+^ and CD8^+^ T cells can further be classified by their expression in terms of CD95, CD44, and CD62L into naïve T cells, stem cell memory T cells (SCM), central memory T cells (CM), and effector memory T cells (EM, 5–8). (**B**) CD4^+^ T cells express higher CD3 levels compared to CD8^+^ T cells. (**C**) Sca-1 expression in CD4^+^ and CD8^+^ T cells. (**D**) Location of T cells (blue) in the FSC/SSC plot (1) and the CD45/SSC plot (2). Source of the plots: (**A**1–4) Panel 1; (**A**5–8,**B**,**C**) Panel 3.

#### 3.1.2. B Cells

B cell gating is shown in Figure 4. These cells can be specifically labeled with CD19 (Figure 4A/2) [34,35,36]. Another good marker for pan-B cells in mice is B220, which was also reported in dendritic cells [37,38]. During early development in the bone marrow, B cells start to express IgM and, as soon as they enter the bloodstream, they begin to co-express IgD (Figure 4A/4) [39,40]. They do not, as yet, produce any specific antibody, but express high MHCII (Figure 4A/2) [41]. After activation via IgD or IgM, B cells develop into B memory or plasma cells, thus producing high amounts of soluble IgM molecules and, later, IgG or IgE [42,43,44]. Memory B cells express CD27, but it is known that CD27^-^ memory B cells also exist [45]. In addition, they express IgG on their surface to be able to react immediately if a known antigen binds again. A marker to identify plasma cells is CD138 (Syndecan-1), which is expressed on different B maturation stages [46,47]. Weisel et al. present a detailed analysis of mouse and human B memory cells and summarize the complexity of the classification [45]. The majority of circulating B cells are antigen-naïve, mature IgM^+^IgD^+^IgG^−^CD27^−^CD138^−^ cells (Figure 4A/3–5). Most of the B cells are Sca-1^+^, as they are still able to differentiate and divide (Figure 4A/6). FSC^high^Sca-1^high^ cells can be found frequently (Figure 4A/6), which might be B cells about to divide into memory or plasma cells. Those are mainly found in lymph nodes, bone marrow, or infected tissues [9]. Therefore, we did not add these markers to our panels.

B cells are located in the SSC^low^ population in the FSC/SSC plot (Figure 4B/1) and in the CD45^high^SSC^low^ population in the CD45/SSC plot (Figure 4B/2).

**Figure 4 cells-13-01583-f004:**
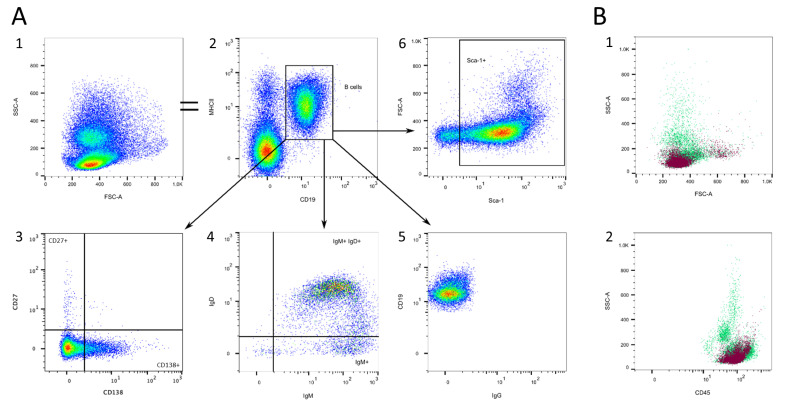
B cell gating. (**A**) B cells can be marked with CD19 (2) and are MHCII^+^ IgM^+^ IgD^+^IgG^−^ (2,4,5). Most B cells express Sca-1 (6). Some of the B cells express CD138 or CD27 (3). (**B**) Location of B cells (violet) in the FSC/SSC plot (1) and the CD45/SSC plot (2). Source of the plots: (**A**) Preliminary experiments; (**B**) Panel 1.

#### 3.1.3. Natural Killer (NK) Cells

Markers for NK cells are mouse strain dependent. The most reliable marker for *C57BL/6* mice is NK1.1 (*NKR-P1*, Figure 5A/2), an activating pan-NK cell receptor, not expressed in other mouse strains, like *BALB/c* [48]. For other mouse strains, the activation marker NCR1 (Nkp46) might also be used [49]. Additionally, NK cells are characterized by the absence of CD3, which distinguishes them from natural killer T (NKT) cells (Figure 5A/2) [8,50]. During maturation, NK cells increase their surface density of CD11b and decrease the expression of CD27, categorizing them into four maturation stages: CD11b^low^CD27^low^, CD11b^low^CD27^high^, CD11b^high^CD27^high^, and CD11b^high^CD27^low^ (Figure 5A/3) [51,52]. The CD11b^high^CD27^high^ NK cell subset was reported to exhibit the highest capacity for cytotoxicity and cytokine secretion [51,53]. Another possibility to analyze NK cell activation is to measure the surface receptor densities of activating (e.g., NK1.1, NCR1, NKG2D, Figure 5A/4 + 5) or inhibitory NK cell receptors (e.g., NKG2A) [49,54,55]. Mature NK cells express CD49b, which is also found in basophils [56,57]. Abel et al. characterized NK cell receptors at different maturation stages [55].

Natural killer T (NKT) cells are a sublineage of T cells that share the characteristics of conventional T cells and natural killer (NK) cells, bridging innate and adaptive immunity [58]. They are positive for CD3, as well as the NK cell marker NK1.1 (Figure 5A/2), but express neither CD4 nor CD8 and should be distinguished from NK cells [59].

NK cells are located in the SSC^low^ population in the FSC/SSC plot (Figure 5B/1), and in the CD45^high^SSC^low^ population in the CD45/SSC plot (Figure 5B/2).

**Figure 5 cells-13-01583-f005:**
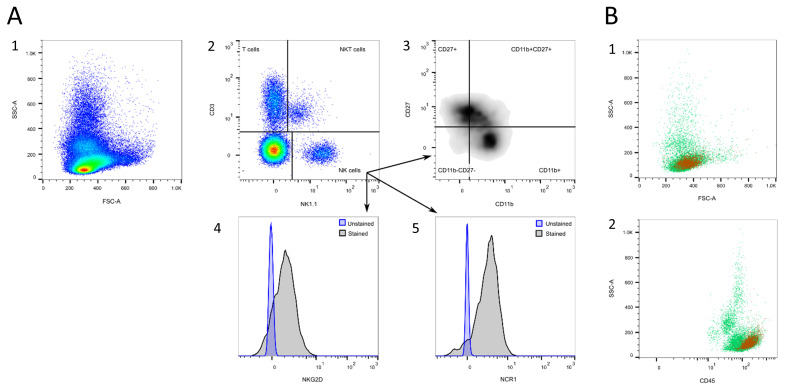
NK cell gating. (**A**) NK cells are characterized by the expression of NK1.1 and the absence of CD3, while NKT cells express NK1.1, as well as CD3 (2). NK cells can be further subdivided into four maturation stages: CD11b^low/−^D27^low/−^, CD11b^low/−^CD27^high/+^, CD11b^high/+^CD27^high/+^, and CD11b^high/+^CD27^low/−^ (3). The activity of NK cells can be analyzed by their expression in terms of activating NK cell receptors, as exemplarily shown for NCR1 and NKG2D (4+5). (**B**) Location of NK cells (orange) in the FSC/SSC (1) and the CD45/SSC plot (2). Source of the plots: Panel 2.

#### 3.1.4. Monocytes and Macrophages

Monocyte gating is shown in Figure 6. Gating monocytes in a blood sample is quite complex, as there is no specific marker. A complete analysis of all the subsets in the mouse spleen led to the classification of CD11b^+^Ly6G^−^CD11c^low^MHCII^−^Ly6C^(+)^ cells as monocytes [60,61,62]. Because NK cells and some B cells also express CD11b, it is important to exclude them first (Figure 6A/2) [51,63,64]. Furthermore, monocytes need to be separated from granulocytes, which also express CD11b [65]. Ly6G can help in excluding neutrophils, which make up the majority of granulocytes (Figure 6A/4) [66]. Additionally, with the help of SSC, monocytes can further be distinguished from eosinophils (Figure 6A/5) [60,61]. To exclude basophils, CD200R3 is necessary (Figure 6A/6) [67]. Monocytes can be further divided into inflammation-promoting Ly6C^high^ and more regenerative Ly6C^low^ populations (Figure 6A/7) [8,68]. Within the monocyte population, there are subpopulations of monocyte-derived dendritic cells characterized by the absence of Ly6C and the presence of MHCII. These cells exhibit dendritic cell-like morphology and are involved in executing various immune defense functions (Figure 6A/7) [62,69].

Monocytes can further develop into macrophages that express F4/80, MHCII, and Siglec-F, but not Ly6G (Figure 6B/1–3) [41,70,71]. The majority of macrophages are located in the tissue, but a small amount also circulates in peripheral blood. They are SSC^high^ and appear as a separated CD11b^high^ population, which helps distinguish them from monocytes and eosinophils (Figure 6B).

In the FSC/SSC plot, monocytes are located in the SSC^low^ population together with lymphocytes (Figure 6C/1). In the CD45/SSC plot, they are also located in the CD45^high^SSC^low^ population (Figure 6C/2). Macrophages are found in the SSC^high^ population and express intermediate CD45 (Figure 6D).

**Figure 6 cells-13-01583-f006:**
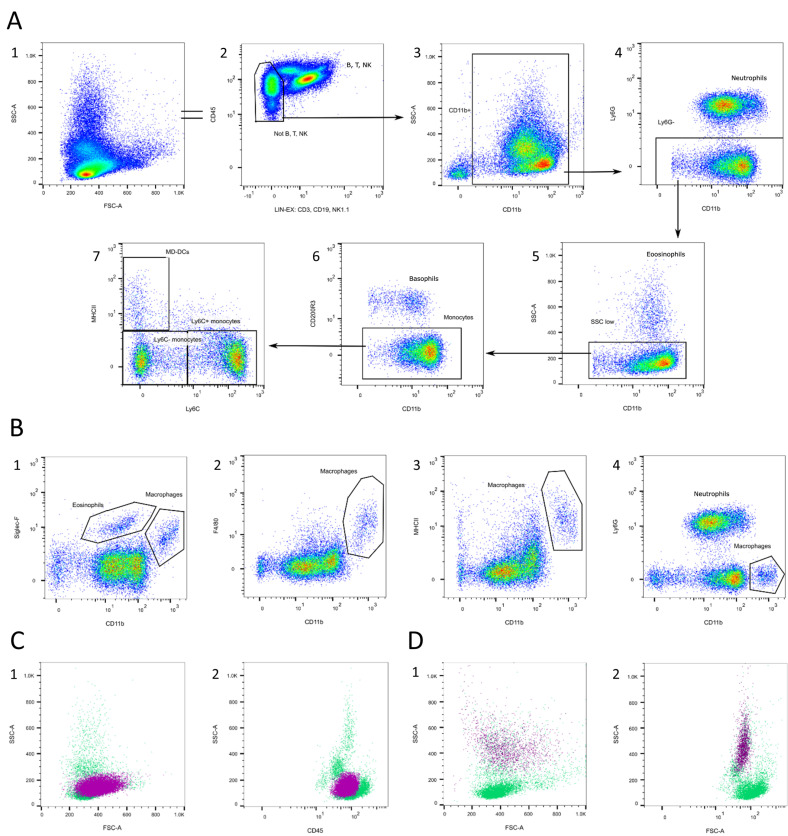
Monocyte and macrophage gating. (**A**) T, B, and NK cells are excluded in a lineage channel (2), then CD11b^−^ and CD11b^high^ cells are excluded (3). Neutrophils are excluded via Ly6G (4), eosinophils via SSC (5), and basophils via CD200R3 (6). Monocytes can further be subdivided into a Ly6C^high/+^ and a Ly6C^low/−^ population (7). Ly6C^−^ MHCII^+^ monocyte-derived dendritic cells can be found (7). (**B**) Macrophages appear as a CD11b^high^ Siglec-F^+^ F4/80^+^ MHCII^+^ Ly6G^−^ population (1–4). (**C**) Location of monocytes (pink) in the FSC/SSC plot (1) and the CD45/SSC plot (2). (**D**) Location of macrophages (pink) in the FSC/SSC plot (1) and the CD45/SSC plot (2). Source of the plots: (**A**,**B**3–4,**C**,**D**) Panel 4; (**B**1) Panel 6; (**B**2) preliminary experiments.

#### 3.1.5. Granulocytes

The gating of granulocyte populations is exemplarily shown in Figure 7. Granulocytes can be classified as neutrophils, eosinophils, and basophils. They express varying levels of CD11b, which necessitates their differentiation from monocytes (Figure 7A/3) [65]. Neutrophils can be determined with high Ly6G and intermediate amounts of Ly6C (Figure 7A/4), which both bind to the anti-granulocyte receptor-1 (Gr-1) [66]. With the help of Siglec-F, eosinophils can be distinguished from monocytes (Figure 7A/5) [61,72,73,74]. Eosinophils express no Ly6C (Figure 7A/5). In some cases, they express the macrophage marker F4/80, but can be distinguished from macrophages with the help of CD11b (see Section 3.1.4) [61,75]. Basophils are identifiable by the expression of CD200R3 (Figure 7A/6 + 7), which might be decreased after activation [67]. Because their CD11b expression spans from negative to low positive, they should not be gated out of the CD11b^+^ population. (Figure 7A/6). They also express the mast cell markers FcεRIα and CD49b, which are found in NK cells, but not in Ly6C cells (Figure 7A/7) [56,57].

Interestingly, not all granulocytes are equally granular. Figure 7B shows where to find the different granulocyte populations in the FSC/SSC and CD45/SSC plots. Basophils are the least granular (Figure 7B/5 + 6), followed by neutrophils (Figure 7B/1 + 2), and highly granular eosinophils (Figure 7B/3 + 4). This makes it possible to roughly subdivide them in the CD45/SSC plot, as basophils also express the least CD45 (Figure 7C). Still, using specific markers is more accurate and, at the very least, CD11b^high^ macrophages should be excluded first.

**Figure 7 cells-13-01583-f007:**
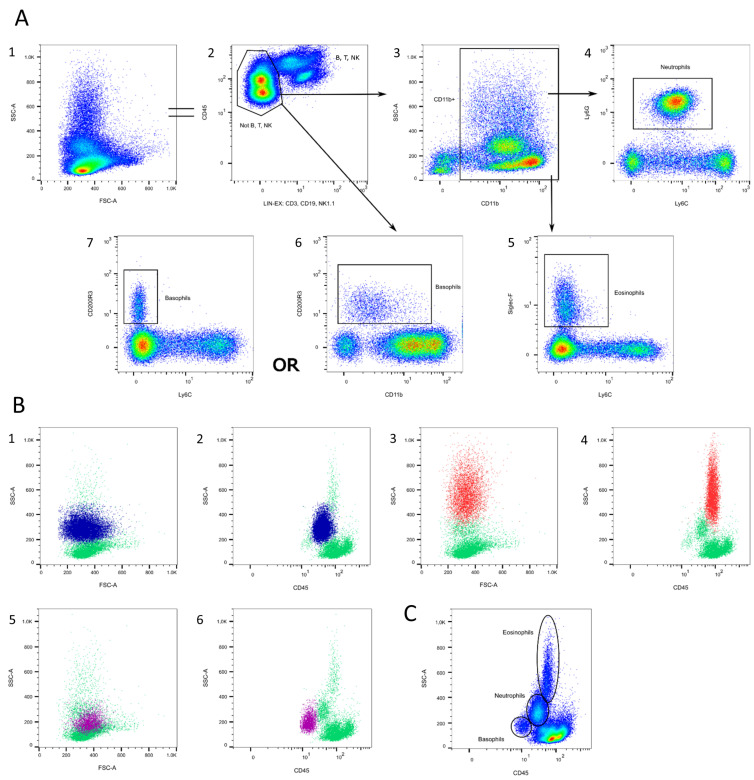
Granulocyte gating. (**A**) T, B, and NK cells are excluded in a lineage channel (2). Neutrophils can be determined as a CD11b^+^ Ly6G^high/+^ Ly6C^int^ population (3 + 4). Eosinophils are characterized as CD11b^+^ Siglec-F^+^ Ly6C^−^ (3 + 5). Basophils are CD11b^int^ CD200R3^+^Ly6C^−^ (6 + 7). (**B**) Location of neutrophils (blue), eosinophils (red), and basophils (pink) in the FSC/SSC plot (1,3,5) and the CD45/SSC plot (2,4,6). (**C**) After excluding CD11b^high^ macrophages, granulocytes can roughly be subdivided in the CD45/SSC plot. Source of the plots: Panel 6.

#### 3.1.6. Dendritic Cells (DCs)

Mouse dendritic cells can be classified as plasmacytoid DCs (pDCs) and conventional DCs (cDCs) [38,76]. Moreover, pDCs are primarily located in the blood and lymphoid tissue and express PDCA-1, B220, and Siglec-H, as well as intermediate levels of MHC II and CD11c (Figure 8A/3 + 4) [77,78]. B220 is also expressed by B cells and is, therefore, not as specific as PDCA-1 or Siglec-H [37,79]. CDC express higher levels of MHC II and CD11c compared to pDCs and can only be found in lymphoid and non-lymphoid tissues [38,41].

Concerning the FSC/SSC plot, pDCs are located next to the SSC^low^ population, but they are slightly bigger and more granular than lymphocytes (Figure 8C/1). In the CD45/SSC plot, they are located on the upper edge of the CD45^high^SSC^low^ population (Figure 8C/2).

**Figure 8 cells-13-01583-f008:**
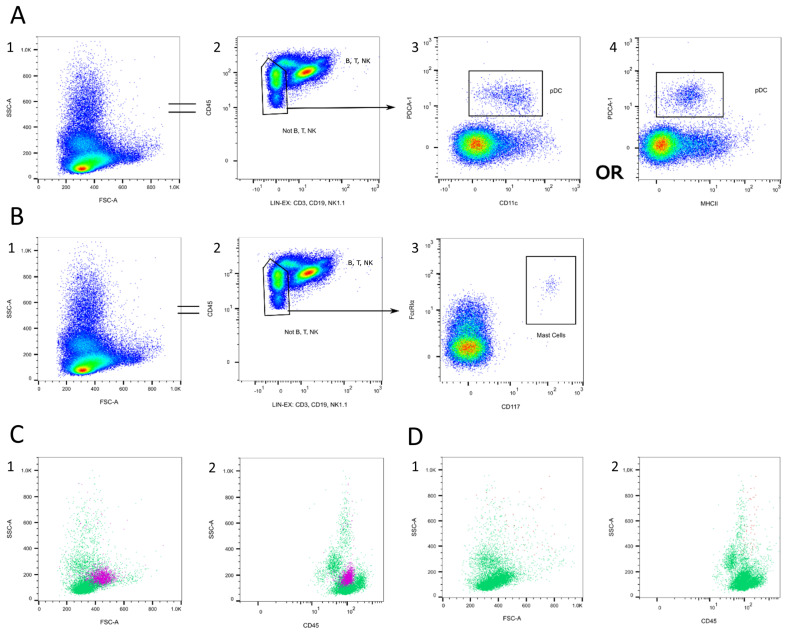
Dendritic cell and mast cell gating. (**A**) T, B, and NK cells are excluded in a lineage channel (2). PDC are characterized as PDCA-1^+^ CD11c^int^ MHCII^int^ (3 + 4). (**B**) T, B, and NK cells are excluded in a lineage channel (2). Mast cells can be determined as FcεRIα^+^CD117^+^ (3). (**C**) Location of DCs (pink) in the FSC/SSC plot (1) and the CD45/SSC plot (2). (**D**) Location of mast cells (red) in the FSC/SSC plot (1) and the CD45/SSC plot (2). Source of the plots: Panel 5.

#### 3.1.7. Mast Cells

Mast cells have a hematopoietic origin and can only be found as rare, immature, precursor cells in the peripheral blood, before migrating into the tissue [80,81]. When circulating in the peripheral blood, they express the stem cell antigen c-kit (CD117) and the IgE receptor FcεRIα (Figure 8B/3) [80,82]. Because FcεRIα is also found in basophils, both antibodies are necessary to identify the mast cell population [80].

Mast cells are highly granular and, therefore, are found in the SSC^high^ population (Figure 8D/1 + 2).

### 3.2. Orientation Values

The presented method and gating strategies were applied to obtain orientation values in the context of age and sex. For this purpose, Panels 1–7 were applied to 3-month-old and 24-month-old *C57BL/6J* mice. Figure 9 shows the obtained results from Panels 1–6 and can be used as orientation values. It includes values for all the described leukocyte subsets, as well as those subtypes that displayed differences between the four groups. An overview of the ranges can be found in the Appendix A. Values that did not show large differences were not considered relevant enough to be presented in the main body of this paper. However, they can be found in the Appendix A.

Lymphocytes make up the majority of all leukocytes with around 60-85%. Most of these are B cells, comprising 35–60% of the leukocytes (Figure 9A). T cells constitute approximately 7–30% of all leukocytes. Males have a lower T cell count than females in the 3-month-old group and older animals display fewer T cells than younger animals (Figure 9B). Large differences could be observed concerning the proportions of CD4^+^ and CD8^+^ T cells, with 45–60% CD4^+^ T cells in the younger mouse group and 15–40% in the older mouse group (Figure 9C). This effect is reversed for CD8^+^ T cells. Younger mice have 30–45% CD8^+^ T cells and aged mice have 40–80% (Figure 9D). Furthermore, large discrepancies were present concerning the T cell maturation stages. The younger cohort showed significantly elevated frequencies of naïve cells (Figure 9E,F) and reduced proportions of the effector memory (EM) and central memory (CM) subsets in both CD4^+^ and CD8^+^ T cells (Figure 9G–J). Additionally, younger mice displayed a slightly higher quantity of T_SCM_ than the aged mouse group (Figure 9K,L). Older mice had marginally higher frequencies of double-negative CD4^−^CD8^−^ T cells than younger mice (Figure 9M). Because the plots of the T cell maturation stages appear highly different between the young and the aged mouse group, they are displayed in the Appendix A. Young females had fewer Sca-1^+^CD4^+^ T cells than males, but the inverse was found in the aged mouse group (Figure 9N). Interestingly, the fraction of Sca-1^+^CD8^+^ T cells increased markedly in the aged mouse group (Figure 9O). CD25^+^CD4^+^ T cells represent around 5–15% of CD4^+^ T cells and did not show large discrepancies between the groups (Figure 9P). The percentage of NK cells in the peripheral blood was 1–8% and NKT cells compromise less than 6%. The aged mouse group displayed fewer NK cells, as well as NKT cells, than the young mice (Figure 9Q,R). The monocyte population accounted for 7–25% of the circulating leukocytes, with slightly more in the aged mouse group (Figure 9S). Macrophages constituted 0–6% of the leukocytes, with the greatest dispersion found in the young female group (Figure 9T). Dendritic cells and mast cells were quite rare in the peripheral blood, with pDCs only making up around 0–1.5% of all leukocytes and mast cells only 0–0.2% (Figure 9U,V). Around 7–30% of the circulating leukocytes were granulocytes. Most of the granulocytes were neutrophils, with a total of 5–25% of all leukocytes. Females appeared to have slightly more neutrophils than males and aged mice have higher percentages than young mice (Figure 9W). The frequencies of eosinophils and basophils do not differ significantly. They are rare in peripheral blood, with values of only 1–4% eosinophils and 0–2% basophils (Figure 9X,Y).

**Figure 9 cells-13-01583-f009:**
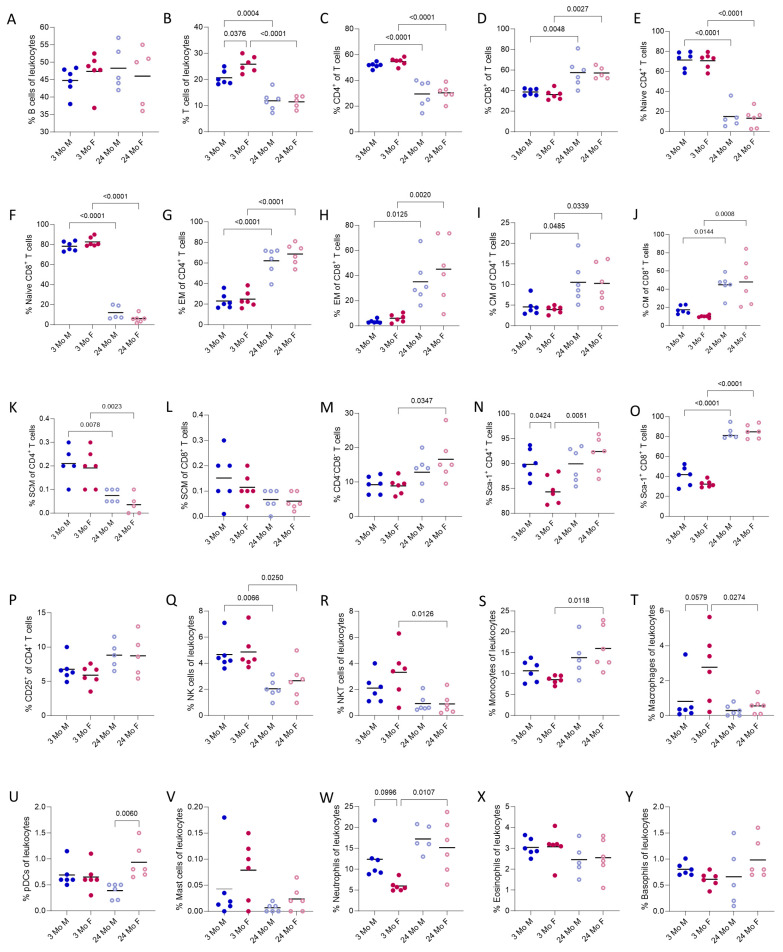
Results from analyzing the blood from a cardiac puncture (Panel 1–6) to be used as orientation values (**A**–**Y**). The values for every leukocyte subset, as well as those subtypes that displayed differences between the groups, are presented. The data are presented as individual values and the mean. Groups: 3-month-old males (3 Mo M), 3-month-old females (3 Mo F), 24-months-old males (24 Mo M), and 24-month-old females (24 Mo F). Brackets indicate *p* values if lower than 0.1.

### 3.3. Appearance of the FSC/SSC and CD45/SSC Plots

The leukocyte types can differ in size and granularity, depending on the individual, their health condition, and the method of blood reprocessing. This leads to an individual appearance in terms of the FSC/SSC plot and the CD45/SSC plot. Representative examples of the FSC/SSC plot can be found in Appendix A.

### 3.4. Comparison of Cardiac and Facial Vein Blood Collection

Figure 10 presents the gating strategy for rough gating of the leukocyte subsets without specific markers with Panel 7, as described in the previous chapters. It must be said that this gating strategy is not as robust as using specific markers, but provides an overview. This is also strengthened by the fact that small populations of cells remain unidentified (Figure 10/6).

Figure 11 shows the results from Panel 7, which was conducted with blood from the facial vein. Both values from the heart blood and facial vein blood for each mouse are linked. The trend remained the same and the results were comparable, but they were still not identical. For B cells, there was a tendency towards higher percentages in the periphery, while the inverse was true for NK cells.

**Figure 10 cells-13-01583-f010:**
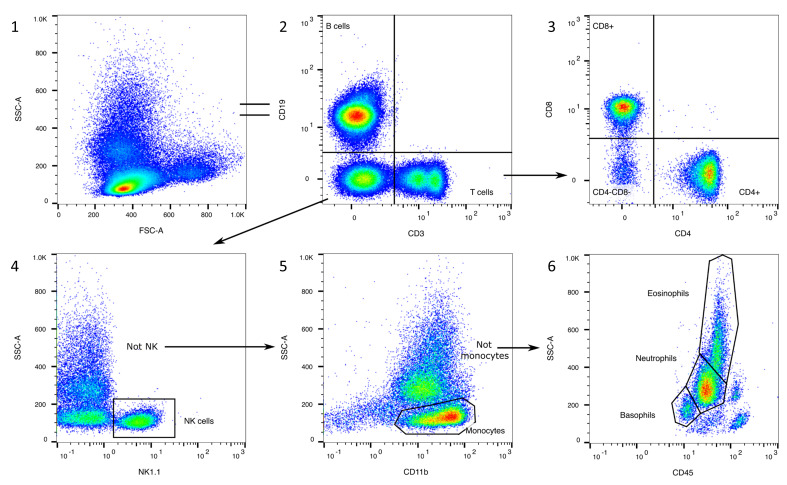
Gating strategy for Panel 7. B cells are identified with CD19 (2). T cells are CD3^+^ (2) and can be divided into CD4^+^, CD8^+^, and CD4^−^CD8^−^ populations (3). After excluding B and T cells, NK1.1^+^ NK cells can be distinguished (4). After excluding NK cells, monocytes can be roughly outlined with the help of CD11b and SSC (5). After excluding monocytes, different granulocyte populations can be roughly categorized by using SSC and CD45 (6).

**Figure 11 cells-13-01583-f011:**
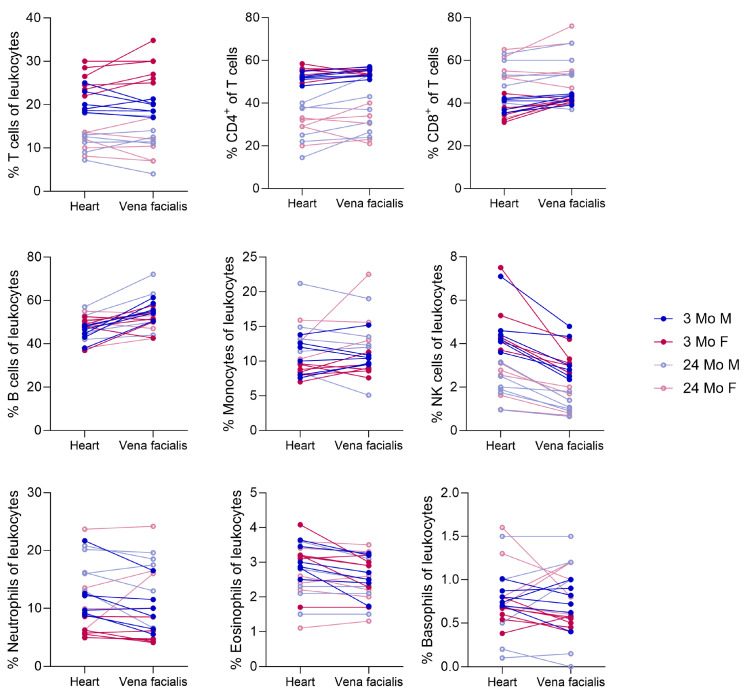
Differences between cardiac (Panels 1–6) and facial vein blood sampling (Panel 7). The lines connect those values that belong to the same mouse.

### 3.5. Validation Approaches

#### 3.5.1. Stress Handling and Blood Glucose Measurements

The experiments took place in a silent environment. The mice were transferred in large beakers, which they entered voluntarily, and were not grabbed by their tail or fixated, if avoidable. As was appropriate, the experimenters acted as calmly and quietly as possible. There were never more people in the room than necessary. Measurements of the blood glucose levels were conducted to ensure that the procedure did not cause unnecessary stress. Dungan et al. describe stress hyperglycemia as a random blood glucose level > 11.1 mmol/L (approx. 2-fold increase), without evidence of previous diabetes in human patients [83]. Stress hyperglycemia was already described in obese *C57BL/6J* mice and normal physiological random blood glucose levels have been measured in different studies and should range from around 7 mmol/L +/− 4 mmol/L [84,85,86]. Therefore, the threshold for stress-induced hyperglycemia was set to 11 mmol/L in our study. Figure 12 shows the results from the blood glucose analyses. Only two animals surpassed the threshold, but did not show deviating values concerning their blood count. Aged mice appeared to have lower blood glucose levels than the young mouse group.

#### 3.5.2. Blood Smears

Blood smears were conducted to validate the results from the flow cytometry. Pictures and results from the blood smears can be found in the Appendix A.

**Figure 12 cells-13-01583-f012:**
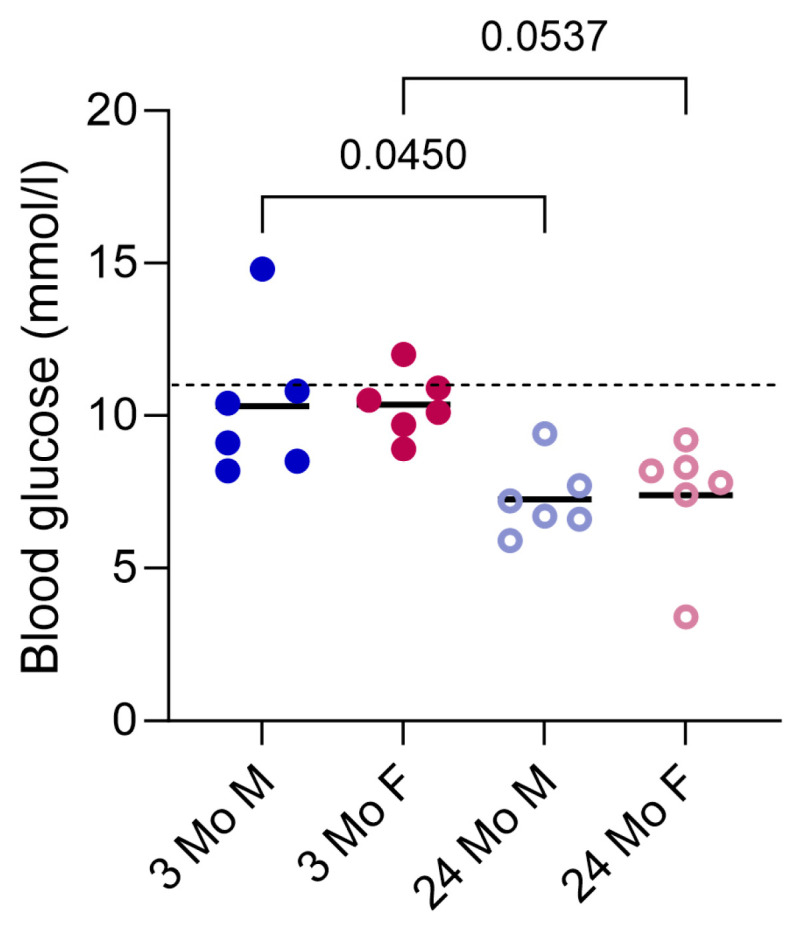
Blood glucose values of all mice in the experiment. Two of the animals surpassed the set threshold of 11 mmol/L (dashed line) for stress hyperglycemia. Data are presented as individual values and the mean. Groups: 3-month-old males (3 Mo M), 3-month-old females (3 Mo F), 24-month-old males (24 Mo M), and 24-month-old females (24 Mo F). Brackets indicate *p* values.

#### 3.5.3. Reproducibility of the Results

Some of the used antibodies can be found in more than one panel and conjugated to different fluorochromes, consistently leading to the same results. In addition, validation experiments were performed, with the blood from the same mouse being divided into several samples and processed individually. Still, the results were comparable. However, changes in the way the blood reprocessing was carried out (e.g., centrifugation steps or incubation times of erythrocyte lysis) caused slightly different results concerning the immune cell proportions, but not by more than 10%.

## 4. Discussion

The present project provides an easy and reproducible protocol for the reprocessing of blood for flow cytometric analysis, as well as suitable markers and gating strategies for analyses of the mouse white blood cell differential. Additionally, orientation values are given in the context of age and sex, which are not yet available in the literature.

In this project, we have presented two possible applications. Both are in the sense of the 3Rs of laboratory animal science (Reduce, Replace, Refine). One is the analysis of the whole blood count by analyzing blood from a cardiac puncture at a final time point in the experiment when the mouse is euthanized anyway. This way, more information can be obtained from the animal without causing further pain, suffering, or damage. It should be kept in mind that after cervical dislocation or opening of the mouse, the amount of blood obtained by cardiac puncture is smaller. The other possible application is the analysis of one or more specific leukocyte populations over the course of a disease. For this, blood should be sampled from the facial vein, optionally under light isoflurane anesthesia. Blood sampling from the facial vein is a well-established method that causes little stress to animals and is well-suited for multiple samplings of small amounts of blood. A maximum of 170 µL per side can be collected from a living mouse and a recovery period of two weeks is required before repeating the blood sampling [87]. Up to three panels can then be analyzed. By taking multiple blood samples within the framework of an animal experiment, individuality can be taken into account, whereby only a very low additional load is exerted on the animal. As we have been able to show, the white blood count is quite individual, and so is the progression of many cancer types and diseases. Therefore, a comparison of different time points in different animals is not as robust as comparing different time points in one animal. By applying our method, fewer animals are needed, and the study outcome is improved.

Another advantage of the method is the possibility of saving the plasma. This step may affect the viability of the cells, but allows multiple further measurements to be performed. Right after blood collection, the blood should be centrifuged at RT with 1500× *g*. The plasma can be collected and frozen at −80 °C, then the solid blood components are resuspended with PBS or FCA buffer. The cell suspension is then used to carry out the protocol. Because some cells are inevitably lost when the blood is processed, no absolute cell counts are given in this study. Additionally, we cannot make any statements about the percentage of leukocytes in peripheral blood, as we performed erythrocyte lysis that concentrated on leukocytes. If the erythrocytes are not eliminated, the antibodies might have poor or no staining, and light-scatter parameters cannot be analyzed due to interference by erythrocytes [88].

The method is intended for staining freshly sampled blood cells. It is also possible to fixate the cells after staining and measuring them in the days following, by using a fixable live/dead dye and a fixation kit. The staining stays stable for at least two days and the results are comparable to fresh cells. Fixation and permeabilization for staining intracellular markers are possible, but the experimenter must be aware of changes in the granularity and size of the cells, as well as the poor comparability to fresh cells. In contrast, we would not recommend staining previously (para-) formaldehyde-fixed or cryopreserved cells. Fixation results in the death of all cells, which is why the antibodies cannot bind properly. Cryopreservation leads to a significant loss of viability; since not all leukocytes perish equally, this affects the results.

Of course, it is possible to rearrange the panels or to combine them into a highly complex multi-fluorochrome panel, as suggested by Kare et al. [89]. Especially the application of facial blood sampling and the integration of the method into an ongoing animal experiment would benefit from this. The choice of the fluorochromes must then be made according to the flow cytometer that is used. However, this requires a device with significantly more channels, and the compensation procedure becomes extremely complicated. We, therefore, decided to keep the panel design in our study simple so that it is reproducible and versatile.

Considering the limitations of the method, it must be remembered that the white blood cell differential is only a momentary snapshot of the situation in the organism, which is subject to many influences. One factor that is quite hard to eliminate is stress. We tried to avoid stress as best as we could and controlled the mouse’s blood glucose levels to check for stress hyperglycemia. The experimenter must be aware that their actions influence the animals and, thus, also the results. If the blood sampling takes place at multiple time points, or without anesthesia, less aversive handling methods, like cup handling, as well as training of the mice, should be considered [90]. Interestingly, we could observe differences concerning the blood glucose levels between the young and aged mouse groups. In line with our results, previous studies have also described lower blood glucose levels in aged mice compared to young mice [91,92]. However, this difference is not the subject of this study and, therefore, will not be discussed in detail.

Changes in the blood collection procedure (arterial/venous/mixed blood), the anesthesia or euthanasia protocol, as well as the time of collection, might cause slightly different results. Therefore, it is essential to always perform the blood collection with identical conditions within the experiment. We do not recommend comparing samples taken from different locations, under different circumstances, or at different times of day. The immune system follows a circadian rhythm, which should be considered when setting the time for blood sampling [93]. We could show that reprocessing blood taken from the facial vein or via a heart puncture results in a slightly different outcome. Besides the blood sampling method, the procedure of blood reprocessing can influence the results. Samples standing on ice for a long period before measurement might lower the viability of the cells. Since not every blood cell will perish equally, and some are more stable than others, this can distort the results. We therefore recommend storing unprocessed blood in the EDTA tube at room temperature, rather than storing the stained samples on ice, and measuring the samples as fast as possible after staining.

It is important to use a sufficient amount of EDTA when handling murine blood. It contains significantly more platelets than human blood and, therefore, clots much faster [94,95]. We therefore recommend coating the syringe and cannula with EDTA. If the number of doublets in the samples consistently exceeds 2%, the EDTA amount should be increased in the sample tube or the FCA buffer [96]. Additionally, we would not recommend drawing any conclusions based on the appearance of the FSC/SSC plot (Appendix A). Nevertheless, the blood from all animals displaying an unconventional FSC/SSC plot can still be analyzed using the markers described in this study.

Within the scope of our project, we were able to highlight differences concerning the proportions and maturation stages of different leukocyte subsets between the mouse groups in our study. The largest effects were prominent between the younger and older mouse groups, especially in the T cell compartment. Immune aging is the subject of many research projects and is characterized by functional and structural alterations in the immune system that lead to a decreased ability to fight infections and a higher incidence of cancer [97]. With age, a decrease in regenerative capacity is described [98]. Hematopoietic stem cells upregulate genes that are associated with myeloid specification and decrease genes that mediate lymphoid specificity [99,100]. Together with thymic involution, this leads to a smaller amount of circulating T cells and, concerning T cell subpopulations, to higher proportions of maturate T cells and lower proportions of naïve T lymphocytes [97,101]. As NKT cells and a small amount of NK cells also develop in the thymus, we could also show this effect on NK and NKT cells. In line with this, we could prove that there was an increase in the myeloid compartment in our aged mouse group (monocytes, granulocytes). In addition, the CD4/CD8 ratio of T cells shifts towards CD8^+^ T cells [101]. We could also show that an increase occurred in the Sca-1 expression in CD8^+^ T cells in both sexes in the aged mouse group. In differentiated T cells, Sca-1 is considered a maturation marker [32]. This again indicates immune aging, particularly in the T cell compartment.

Besides age, sex is also a biological variable that affects the functions of the immune system. In our project, females had higher proportions of T cells and macrophages, as well as lower expression of Sca-1 in T cells, in the younger mouse group. The strong increase in Sca-1^+^CD4^+^ T cells in old females is most likely because females are generally able to mount a stronger immune reaction than males and, therefore, also develop a stronger immunological memory [102]. This is mainly due to the higher activation capacity of immune cells, as well as cytokine and chemokine production [102]. Additionally, estrogen was proven to increase hematopoietic stem cell self-renewal [103]. In contrast to this, males had larger frequencies of neutrophils in our study. Still, it could be shown that female neutrophils and macrophages have a higher level of phagocytic activity than males [104]. In summary, we observed differences in the leukocyte fractions in the context of age and sex, which must be considered in future research. Still, it must be emphasized that the orientation values provided in this project are only intended to show what results can be achieved with the described method and are, therefore, discussed rather briefly.

In conclusion, the presented method can be applied to multiple research areas that include laboratory mice and has been proven to be reliable in many types of applications. This allows statements to be made about the status of the immune system, as well as blood diseases and systemic infections. Especially in the progression of many cancer types (e.g., pancreatic cancer, breast cancer), the reaction of the immune system is of particular interest. As blood analyses are frequently used in human medicine, the transferability of the method to humans is very high. The present study fills the gap in the knowledge in terms of the rare information on flow cytometric analysis of mouse blood and, thus, lays the foundation for further investigations in this area. 

## Figures and Tables

**Table 1 cells-13-01583-t001:** Seven panels were used in the project. Panels 1–6 were conducted with blood from the heart puncture and Panel 7 was conducted with peripheral blood from the facial vein.

	VioBlue	VioGreen	FITC/VioBright	PE	PE-Vio770	APC	APC-Vio770
Panel 1: Lymphocytes	CD45	CD8b	CD3	CD25	Sca-1	CD4	CD19
Panel 2: NK cells	CD45	CD11b	CD3	NCR1	CD27	NKG2D	NK1.1
Panel 3: T cells	CD45	CD8b	CD62L	CD95	CD4	CD3	CD44
Panel 4: Monocytes	CD45	MHCII	Ly6C	CD11b	CD200R3	Ly6G	CD3, CD19, NK1.1
Panel 5: Dendritic cells/mast cells	CD45	MHCII	FcεRIα (IgE)	PDCA-1	CD11c	CD117	CD3, CD19, NK1.1
Panel 6: Granulocytes	CD45	Ly6C	Siglec-F	CD11b	CD200R3	Ly6G	CD3, CD19, NK1.1
Panel 7: Peripheral overview	CD45	CD8b	CD3	CD11b	CD4	NK1.1	CD19

## Data Availability

The datasets (FCS Files) that were generated and analyzed in this study can be found at Flow Repository (http://flowrepository.org/) under the repository ID FR-FCM-Z7C9.

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
