# Peer review of "A Flow Cytometry-Based Examination of the Mouse White Blood Cell Differential in the Context of Age and Sex"

_cells, 2024, doi:10.3390/cells13181583_

Round 1

Reviewer 1 Report

Comments and Suggestions for Authors

Please find attached the review report.

Comments on the Quality of English Language

The quality of the English Language is good. Some typos were found and described in the review report.

Author Response

Dear Reviewer 1,

thank you really much for the evaluation of our manuscript. We are pleased that you have rated our work positively. In the following, we will address your comments in detail:

Comment 1: The introduction should cover current state of the research field. Please add more literature reviews on line 46 about other studies on using flow cytometry to study mice blood, for example, Skordos, I., Demeyer, A., & Beyaert, R. (2021). Analysis of T cells in mouse lymphoid tissue and blood with flow cytometry. STAR protocols, 2(1), 100351.

Response 1: Thank you for the suggestion of this publication. It is about mouse T cell subsets and contains a useful protocol for mouse blood analyses that is similar to ours. It additionally includes other immune cell compartments like the thymus and spleen. Still, it does not contain results from applying the method to mice and does not include other leukocyte subsets than T cells. We have added a reference to line 50 in the introduction: “Skordos et al. provide a protocol for the analysis of mouse T cell subsets and analyze them in different immune cell compartments. Despite many different publications on partial aspects of the topic, we are not aware of any literature that deals comprehensively and exclusively with mouse blood analyses and contains achievable results. Therefore, this paper provides an overview of this topic and promotes standardized flow cytometry-based analyses of mouse blood.” Additionally, we have added references to this publication in the T cell chapter.

Comment 2: In Figure 2B, please label the dead cells on plot 1, and erythrocytes and platelets on plot 2. Similarly, label other cell populations if possible on the figures like what the authors did in Figure 3B/4.

Response 2: Thank you for this point. In Figure 2B, we have labeled the dead cells in blue and the erythrocytes and platelets in red and indicated this in the figure legends. We now labeled other cell populations that can be identified clearly in Figures 3, 6, 7, and 8. We cannot label all the populations, as they often include several populations, and therefore cannot be identified clearly. Please let us know if something is still missing to be labeled.

Comment 3: In Figure 9, please rearrange the order of the subplots, so they are in the same order with what is described in the text from line 421 to line 453. This would be helpful for the reader to read through Figure 9, and don’t need to go back and forth.

Response 3: This is a really good point, that we implemented in Figure 9.

Comment 4: On Figure 9H, there is a data point for 24 Mo F that is around 18%, which is around 10% higher than the rest of the data point, is there any chance that it is an outlier? If it is an outlier, then would removing it lower the mean value for this group, and will it lead to a significant difference between this group and the 3 Mo F group? Please discuss it.

Response 4: Thank you for this question. Excluding this data point lowers the mean from 8.02 to 6.0 and changes the p-value from 0.84 to 0.27. The p-values for the tests of the other groups also decrease, but none of them surpasses the limit of 0.1. Therefore, no significance can be reached by excluding this value.

We decided on only excluding the outliers of those mice where we saw a biological reason for a deviation from physiology. This was not the case for this individual, which is why it is included in the analysis. Since we only used a small n-count in our projects, it is difficult to name a deviating value as an outlier without further reasons. We cannot say whether the distribution of the results would change with a larger number of mice used. Unfortunately, doing this would not be in the sense of the 3R. These values might look slightly different for every mouse strain or due to external influences. Every researcher should compare his results to his own control group.

Comment 5: In section 3.3, the authors only discussed the difference between the location of the blood collection. Please also add discussions on the differences between the age and sex.

Response 5: Thank you very much for this point. Unfortunately, we have not clearly separated the results and the discussion here. Section 3.3 should only compare the results from the different locations and was not meant to include a discussion on this. To separate this more from the discussion, we removed the sentence: “It could be shown that changes in the blood collection procedure (location and time of collection) might cause slight differences in the immune cell proportions.“ from this section, as there is already a paragraph on this in the discussion section. The differences between age and sex are presented in chapter 3.1 and considered in the discussion section.

Due to the same reasons, we have also moved the following paragraph from chapter 3.2: “We would not recommend making any statements about the status of the immune system based on the appearance of the FSC-SSC-plot. Nevertheless, the blood of all animals in which the FSC-SSC-plot looks unconventional can still be analyzed using the markers described above.” and added it to the discussion section: “Additionally, we would not recommend drawing any conclusions based on the appearance of the FSC-SSC plot. Nevertheless, the blood of all animals displaying an unconventional FSC-SSC plot can still be analyzed using the markers described in this study.”

All discussions on the topics raised by you are now bundled together in the discussion section.

Comment 6: On line 567, the authors mentioned limitation of stress, which has already been discussed in Section 3.4.1. Please rearrange these two sections, would recommend to move the text from line 576 to line 579 “One factor that is quite hard to eliminate, … should be considered. [91]” to section 3.4.1 on page 16.

Response 6: Thank you very much for pointing this repetition out. Introduction, results and discussion are not clearly separated here. As chapter 3.4.1 is part of the results section, it should only be outlined, how we tried to minimize and control stress in our project. Therefore we removed the sentence from this chapter: “Since stress might cause changes in the immune cell proportions or activation status, it is essential to cause as less stress as possible to the animals while blood sampling.“ Also, we moved the following sentence to the introduction: “In acute stress situations, the hypothalamic-pituitary-adrenal axis and the sympathetic nervous system are activated, which results in a release of stress hormones from the liver and therefore stress-induced hyperglycemia.“

Comment 7: On line 47, correct the typo for the number “with 72.4%”.

Response 7: This has been corrected.

Comment 8: For the title of Table 1 on line 153, the first couple of words “Table 1 shows” can be removed to make the sentence more concise.

Response 8: Thank you, we have changed this. We additionally applied the same to the legend of figure 9.

Comment 9: On line 339, correct the typo “were” to “where”.

Response 9: We have corrected this.

Comment 10: On line 632, would it be more accurate to change the phrase “laboratory animal” to “laboratory mice”? Given the concern that there are other lab animals.

Response 10: This is a good point and we have changed this.

We strongly hope that you find our changes acceptable and that you will endorse our manuscript for publication.

Thank you for your time and consideration. Best regards,

The authors.

Reviewer 2 Report

Comments and Suggestions for Authors

The manuscript entitled “A flow cytometry-based examination of the mouse white blood 2 cell differential in the context of age and sex” by Elise et al, described comprehensive flowcytometry-based phenotypic analysis to decipher the characteristic change of immune cell populations in the context of mice age and sex. They have further discussed the underlying factors contributing to those cell population properties. The light point of this manuscript is the very detailed analysis of the flow cytometry plot and it did make a good quality experimental article. Furthermore, this topic is of broad interest to the readership of cells. Thus, I recommend the manuscript be published as it is.

Author Response

Dear Reviewer 2,

Thank you very much for your thoughtful and encouraging feedback on our manuscript.  We are grateful for your positive evaluation and are pleased to hear that you found the detailed analysis of the flow cytometry data to be a strong point of our work.

Your recommendation for publication without revisions is greatly appreciated, and we are glad that you believe our research will be of broad interest to the readership of Cells.

Thank you again for your time and support.

Best regards,

The authors.

Reviewer 3 Report

Comments and Suggestions for Authors

This is a very interesting work about a flow cytometry based examination of the mouse white blood cell differential in the context of age and sex.

The authors should try to express in a more English way. Some sentences, especially in the introduction and the methods, need rephrasing, putting the words in a different order, more English-like.

Some parts of the methods especially long explanations in the gating strategies, like the definition of the different subsets, could be mentioned in the introduction. Why is the word reprocessing used instead of processing?

The results are clearly presented. I don't know whether the values obtained from 6 or less mice could be referred as reference values, perhaps they could be named otherwise.

The discussion is very long. Most of it contains the authors' thoughts and not a comparison with other publications, like e.g the paper by Cossarizza et al, mentioned in the introduction.

In general the authors present a very useful approach with a lot of details, to be applied in every laboratory.

Comments on the Quality of English Language

The English needs a thorough check, some sentences are not written in a correct English.

Author Response

Dear Reviewer 3,

Thank you very much for taking the time to review our manuscript. We are pleased that you have rated our work positively. In the following, we will address your comments in detail.

Comment 1: The authors should try to express in a more English way. Some sentences, especially in the introduction and the methods, need rephrasing, putting the words in a different order, more English-like.

Response 1: Thank you very much for pointing this out to us. To address this point carefully, we consulted a Native Speaker to supervise the manuscript and changed the structure in the following sentences:

  • Line 14-15: “Regarding the mouse blood count, there is a lack of information.” → “However, there is a lack of information regarding the mouse blood count.”
  • Line 16-17: “Furthermore, we present two possible applications: ….” → “We also present two further possible applications: …“
  • Line 43: “… gating strategies, as well as how to avoid frequent problems …” → “… gating strategies, and ways to avoid frequent problems …“
  • Line 44 – 46: “In contrast to that, we find very little information about the flow cytometry-based blood count of mice, although the mouse is the most frequently used mammal in research, with 72.4% of all vertebrates used in Germany in 2022.” → “In contrast, very little information exists about the flow cytometry-based blood count of mice despite the mouse being the most frequently used mammal in research. In Germany, mice constituted 72.4% of all vertebrates used in 2022.“
  • Line 63-64: “This means that more than 50% of all blood leukocytes are granulocytes.” → “This means that granulocytes constitute more than 50% of all blood leukocytes.”
  • Line 73-74: “Age, sex, gravidity, lactation, feeding regime, environmental influences but also stress can have an impact …” → “Age, sex, gravidity, lactation, feeding regime, environmental influences, and even stress can have an impact …”
  • Line 87-90: “In this case, the analysis can also be integrated into an ongoing experiment at several time points using peripheral blood from the facial vein, to show changes in peripheral blood leukocytes during the progression of a disease.“ → “In this case, the analysis can also be integrated into an ongoing experiment at several time points. Peripheral blood from the facial vein can be used to show alterations in blood leukocytes during the progression of a disease.“
  • Line 169-172: “The figures 2-8 include representative plots from the young mouse group, as well as some preliminary experiments to best demonstrate the gating strategy, and to prove the statements that will be made in the following chapters.“ → “The figures 2-8 include representative plots from the young mouse group. Also included are some preliminary experiments to best demonstrate the gating strategy, and to prove the statements that will be made in the following chapters.“
  • Line 180-181: “There is no clear separation between lymphocytes and monocytes, as we know it from human blood.“ → “There is no clear separation between lymphocytes and monocytes, as is the case with human blood.“
  • Line 227-228: “Therefore FoxP3 is more specific, but as it is an intracellular transcription factor, it is not used in this study.” → “Therefore, FoxP3 is more specific, but it is not used in this study as it is an intracellular transcription factor.“
  • Line 236-238: “T cells are located in the SSClow population of the FSC-SSC-plot (Fig. 3D/1) and express a high amount of CD45, therefore located in the CD45highSSClow population of the CD45-SSC-plot (Fig. 3D/2).” → “… FSC-SSC-plot (Fig. 3D/1). They express a high amount of CD45 and are therefore located …”
  • Line 246-247: “After activation via IgD or IgM, B cells develop into B memory cells or to plasma cells and produce lots of soluble IgM molecules and later, IgG or IgE.“ → “After activation via IgD or IgM, B cells develop into B memory cells or plasma cells, thus producing high amounts of soluble IgM molecules and, later, IgG or IgE.“
  • Line 342-343: “Eosinophils express no Ly6C (Fig. 7A/5), but sometimes the macrophage marker F4/80, but can be distinguished from macrophages with the help of CD11b.“ → “Eosinophils express no Ly6C (Fig. 7A/5). In some cases, they express the macrophage marker F4/80, but can be distinguished from macrophages with the help of CD11b.“
  • Line 345-346: “Because their CD11b expression reaches from negative to low positive, they should rather not be gated out of the CD11b+“ → “Because their CD11b expression spans from negative to low positive, they should not be gated out of the CD11b+ population.“
  • Line 353-354: “Still, using specific markers is more accurate, and at least CD11bhigh macrophages should be excluded first.“ → “Still, using specific markers is more accurate, and at the very least, CD11bhigh macrophages should be excluded first.“
  • 396-397: “The proportions were then compared to the flow cytometry analyses and it could be shown that the results were comparable.” → “The proportions were then compared to the flow cytometry analyses. The results were demonstrated to be comparable.“
  • Line 407: “It is possible to do at least one, maximum of three panels …” → “It is possible to do at least one and up to three panels …”
  • Line 432-434: “Those values that did not show large differences and were therefore considered less relevant to present in the main body, can be found in the supplementary material.“ → “Values that did not show large differences were not considered relevant enough to present in the main body of this paper. However, they can be found in the supplementary material.“
  • Line 435-436: „Most of them are B cells with 35-60% of the leukocytes.” → “Most of these are B cells, comprising 35-60% of the leukocytes.”
  • Line 472: “There are no big differences concerning the frequencies of eosinophils and basophils.“ → “The frequencies of eosinophils and basophils do not differ significantly.”
  • Line 497-498: “Of course, the experimenters acted as calm and quiet as possible and there were not more people in the room than necessary.“ → “As it was appropriate, the experimenters acted as calmly and quietly as possible. There were never more people in the room than necessary.“
  • Line 539: “… by using only one, maximum three panels.” → “A maximum of three panels can then be analyzed.“
  • Line 582-584: “In line with our results, previous studies also described that aged mice have lower blood glucose levels than young mice.” → “In line with our results, previous studies also described lower blood glucose levels in aged mice compared to young mice.“
  • Line 639: “… and are therefore not discussed in detail.” → “… and are therefore discussed rather briefly.”

We have also made minor changes to comma placement, correct spelling, and word choice. We strongly hope that the English style of our manuscript has now improved and fulfills the language standard of  “Cells”.

Comment 2: Some parts of the methods especially long explanations in the gating strategies, like the definition of the different subsets, could be mentioned in the introduction.

Response 2: Thank you for this recommendation. Finding and describing the most suitable markers for identifying and analyzing mouse leukocyte subsets is one of the key points of our manuscript, which is why we describe this extensively. We believe that moving these explanations to the introduction would extend the frame of the introduction. Additionally, most of these markers are supported by the figures in the methods section and linked to them, which is why we cannot separate them. Also, we believe that having the information about markers and gating strategies in one place is helpful for the reader who needs information about a certain leukocyte subset.

Comment 3: Why is the word reprocessing used instead of processing?

Response 3: We use the word reprocessing because we describe a procedure that follows euthanasia, blood sampling, and transferring the sample to an EDTA tube. Also, we describe the possibility of saving plasma before performing the protocol in the discussion, whereby the procedure remains the same. The protocol can therefore be regarded as a continuative procedure.

Comment 4: I don't know whether the values obtained from 6 or less mice could be referred as reference values, perhaps they could be named otherwise.

Response 4: This is a good point. Of course, due to the small n-count, we cannot tell where the limit to pathological alterations should be set. Therefore, we changed this to “orientation values”.

Comment 5: The discussion is very long. Most of it contains the authors' thoughts and not a comparison with other publications, like e.g the paper by Cossarizza et al, mentioned in the introduction.

Response 5: The discussion is indeed very long. We discuss all the advantages as well as the limitations of the method and give suggestions on how to handle them. Additionally, we discuss the results themselves. Much of the content of the discussion is based on personal experience won during the establishment of the method, which we do not wish to withhold from the reader. We are convinced that sharing as much of our experience as possible with the reader will lead to better reproducibility of the method. It is therefore very hard to exclude parts of the discussion, as we consider them all as important. Nevertheless, we have tried to shorten the discussion a little by removing the following sentences:

  • “Of course, other methods of blood collection are also an option, but we consider the method of facial vein blood sampling as the most gentle and safe.“
  • “However, each researcher must decide for themselves which method of blood collection is most suitable for their project.“
  • “Only two mice surpassed the threshold but didn’t show deviating values concerning their blood count.“
  • “Still, it must be pointed out that the blood glucose level is dependent on the last food intake, and the mice were not sober in our study.“
  • “Therefore, the experiments should always be performed after the same protocol.“
  • “This is why the amount of EDTA contained in the sample tube might not be enough, and….”
  • “The method can be applied to any research project involving mice. This allows statements to be made about the status of the immune system as well as blood diseases and systemic infections. Especially in the progression of many cancer types (e. g. pancreatic cancer, breast cancer), the reaction of the immune system is of particular interest.“ This paragraph was moved to the end of the discussion and merged with the conclusion.

Other sentences and paragraphs were rearranged and merged to shorten the text by not losing the content.

The paper by Cossarizza et al. is a very valuable and detailed resource for phenotyping human and murine leukocytes, as it is a guideline for flow cytometry-based analyses. It contains numerous markers and protocols for isolating leukocytes in tissue and partly in blood. Nevertheless, the paper does not focus on blood norincludes obtained results, which is why it is not considered in the discussion. We added a statement on this to the introduction (line 51): “Despite many different publications on partial aspects of the topic, we are not aware of any literature that deals comprehensively and exclusively with mouse blood analyses and contains achievable results. Therefore, this paper provides an overview of this topic and promotes standardized flow cytometry-based analyses of mouse blood.“

We sincerely hope that our revisions meet your expectations and that you will consider recommending our manuscript for publication.

Thank you for your time and support. Best regards,
The authors.